# Should Lymph Nodes Be Retrieved in Patients with Intrahepatic Cholangiocarcinoma? A Collaborative Korea–Japan Study

**DOI:** 10.3390/cancers13030445

**Published:** 2021-01-25

**Authors:** Chang Moo Kang, Kyung-Suk Suh, Nam-Joon Yi, Tae Ho Hong, Sang Jae Park, Keun Soo Ahn, Hiroki Hayashi, Sae Byeol Choi, Chi-Young Jeong, Takeshi Takahara, Shigehiro Shiozaki, Young Hoon Roh, Hee Chul Yu, Takumi Fukumoto, Ryusei Matsuyama, Uyama Naoki, Kazuki Hashida, Hyung Il Seo, Takehiro Okabayashi, Tomoo Kitajima, Sohei SATOI, Hiroaki Nagano, Hongbeom Kim, Kaoru Taira, Shoji Kubo, Dong Wook Choi

**Affiliations:** 1Department of Surgery, Yonsei University College of Medicine, Seoul 03722, Korea; cmkang@yuhs.ac; 2Department of Surgery, Seoul National University College of Medicine, Seoul 03080, Korea; kssuh@snu.ac.kr (K.-S.S.); gsleenj@hanmail.net (N.-J.Y.); surgeonkhb@gmail.com (H.K.); 3Department of Hepatobiliary and Pancreas Surgery, Seoul St. Mary’s Hospital, College of Medicine, The Catholic University of Korea, Seoul 06591, Korea; gshth@catholic.ac.kr; 4Center for Liver Cancer, National Cancer Center, Goyang 10408, Korea; spark@ncc.re.kr; 5Department of Surgery, Keimyung University Dongsan Hospital, Keimyung University School of Medicine, Daegu 42601, Korea; ahnksmd@gmail.com; 6Department of Surgery, Tohoku University Graduate School of Medicine, Sendai 980-8575, Japan; h-hayashi@surg.med.tohoku.ac.jp; 7Department of Surgery, Korea University College of Medicine, Seoul 02841, Korea; csbroad@hanmail.net; 8Department of Surgery, College of Medicine Gyeongsang National University, Jinju 52727, Korea; drjcy@hanmail.net; 9Department of Surgery, Iwate Medical University School of Medicine, Iwate 028-3694, Japan; takahara@iwate-med.ac.jp; 10Department of Surgery, Hiroshima City Hiroshima Citizens Hospital, Hiroshima 730-8518, Japan; shizaki@city-hosp.naka.hiroshima.jp; 11Department of Surgery, Dong-A University College of Medicine, Busan 49201, Korea; gsryh@dau.ac.kr; 12Department of Surgery, Jeonbuk National University Medical School, Jeonju 54907, Korea; hcyu@jbnu.ac.kr; 13Department of Surgery, Kobe University Graduate School of Medicine, Kobe 657-850, Japan; Fukumoto@kobe-u.ac.jp; 14Department of Gastroenterological Surgery, Yokohama City University Graduate School of Medicine, Yokohama 326-0027, Japan; ryusei@terra.dti.ne.jp; 15Department of Surgery, Hyogo College of Medicine, Nishinomiya 663-8501, Japan; uynk@hyo-med.ac.jp; 16Department of General Surgery, Kurashiki Central Hospital, Kurashiki 710-8602, Japan; kh14813@kchnet.or.jp; 17Department of Surgery, Pusan National University College of Medicine, Busan 49241, Korea; seohi71@hanmail.net; 18Department of Gastroenterological Surgery, Kochi Health Sciences Center, Kochi 781-8555, Japan; tokabaya@gmail.com; 19Department of Surgery, Nagasaki Medical Center, Nagasaki 856-0835, Japan; t-kita@juno.ocn.ne.jp; 20Department of Surgery, Kansai Medical University, Osaka 573-1191, Japan; satoi@hirakata.kmu.ac.jp; 21Department of Gastroenterological, Breast and Endocrine Surgery, Yamaguchi University Graduate School of Medicine, Yamaguchi 753-8511, Japan; hnagano@yamaguchi-u.ac.jp; 22Department of Surgery, Dongguk University College of Medicine, Goyang 10326, Korea; 23Department of Surgery, Otsu Red Cross Hospital, Otsu 520-0046, Japan; K.TAIRA@otsu.jrc.or.jp; 24Department of Hepato-Biliary-Pancreatic Surgery, Graduate School of Medicine, Osaka City University, Osaka 558-0022, Japan; 25Department of Surgery, Sungkyunkwan University School of Medicine, Seoul 16419, Korea

**Keywords:** cholangiocarcinoma, lymph nodes, metastasis, nomograms

## Abstract

**Simple Summary:**

Intrahepatic cholangiocarcinoma (IHCC) is the second most common primary hepatic malignant tumor after hepatocellular carcinoma (HCC). The prevalence of lymph node metastases (LNM) detected at surgery for IHCC has been reported as 25–50%, and lymph node metastasis is known to be significantly associated with poor survival outcomes. However, the oncologic value of lymph node dissection in resected IHCC is still controversial. According to the present Korea–Japan international collaborative study, it was found that surgical retrieval of more than four lymph nodes (≥4 LNs) could improve survival outcome in resected IHCC with LNM. Based on preoperatively detectable parameters, a nomogram was established to predict LNM to suggest tailored intraoperative LN management in patients with IHCC. Further prospective research is needed to validate the present surgical strategy in resected IHCC.

**Abstract:**

Background: This study was performed to investigate the oncologic role of lymph node (LN) management and to propose a surgical strategy for treating intrahepatic cholangiocarcinoma (IHCC). Methods: The medical records of patients with resected IHCC were retrospectively reviewed from multiple institutions in Korea and Japan. Short-term and long-term oncologic outcomes were analyzed according to lymph node metastasis (LNM). A nomogram to predict LNM in treating IHCC was established to propose a surgical strategy for managing IHCC. Results: A total of 1138 patients were enrolled. Of these, 413 patients underwent LN management and 725 did not. A total of 293 patients were found to have LNM. The No. 12 lymph node (36%) was the most frequent metastatic node, and the No. 8 lymph node (21%) was the second most common. LNM showed adverse long-term oncologic impact in patients with resected IHCC (14 months, 95% CI (11.4–16.6) vs. 74 months, 95% CI (57.2–90.8), *p* < 0.001), and the number of LNM (0, 1–3, 4≤) was also significantly related to negative oncologic impacts in patients with resected IHCC (74 months, 95% CI (57.2–90.8) vs. 19 months, 95% CI (14.4–23.6) vs. 11 months, 95% CI (8.1–13.8)), *p* < 0.001). Surgical retrieval of more than four (≥4) LNs could improve the survival outcome in resected IHCC with LNM (13 months, 95% CI (10.4–15.6)) vs. 30 months, 95% CI (13.1–46.9), *p* = 0.045). Based on preoperatively detectable parameters, a nomogram was established to predict LNM according to the tumor location. The AUC was 0.748 (95% CI: 0.706–0.788), and the Hosmer and Lemeshow goodness of fit test showed *p* = 0.4904. Conclusion: Case-specific surgical retrieval of more than four LNs is required in patients highly suspected to have LNM, based on a preoperative detectable parameter-based nomogram. Further prospective research is needed to validate the present surgical strategy in resected IHCC.

## 1. Introduction

Intrahepatic cholangiocarcinoma (IHCC) is the second most common primary hepatic malignant tumor after hepatocellular carcinoma (HCC) [1,2]. IHCC represents about 10–20% of all primary liver cancers [3]. Recently, the incidence of IHCC has also been reported to be increasing worldwide [4].

Surgical resection is the mainstay of curative-intent treatment for IHCC and is associated with improved survival in selected patients. In 1997, the Liver Cancer Study Group of Japan proposed that regional lymph nodes of IHCC should be categorized into three groups based on lymph node mapping studies [5]. Recently, the 8th American Joint Committee on Cancer (AJCC) Cancer Staging Manual also suggested that left-sided IHCC should include inferior phrenic, hilar, and gastrohepatic lymph nodes as regional lymph node groups. For right-sided IHCC, regional lymph nodes include hilar, periduodenal, and peripancreatic lymph node areas [6]. These regional lymph nodes are theoretically surgical targets that should be dissected during curative surgical procedures.

The prevalence of lymph node metastases detected at surgery for IHCC has been reported as 25–50% [7]. The literature suggests that lymph node metastasis is significantly associated with poor survival outcomes in patients who undergo hepatic resection for IHCC [8,9]. However, the oncologic value of lymph node dissection in resected IHCC is still controversial [10,11].

Since 2015, Korea and Japan have launched the first mutual collaboration study in the field of Hepato-Biliary-Pancreatic surgery. Several important achievements have already been published, and some of them are being processed for future publications [12,13,14]. The aim of the present study is to investigate the oncologic role of lymph node management in treating IHCC, based on a large-volume study population, and to propose a potential surgical strategy for managing lymph nodes when resecting IHCC.

## 2. Results

### 2.1. General Characteristics of Patients and Primary Tumors

A total of 828 and 310 patients with resected IHCC were enrolled in Korea and Japan, respectively. There were 758 male patients (66.6%) with an average age of 63.4 years. Five hundred and fifty-eight patients (50.4%) had right-sided tumors. Left hemihepatectomy was the most common procedure performed in 325 patients (28.6%, Table 1).

Most patients underwent combined caudate lobectomy (934 patients, 82.4%). Seven hundred and twenty-five patients (63.7%) underwent surgical management for lymph node retrieval. Combination resection of the diaphragm was the most common procedure, excluding the gallbladder (Appendix A).

Mean pathologic tumor size was 5.1 ± 3.0 cm, and the mean number of retrieved lymph nodes was six (5.9 ± 8.3). The mean number of metastatic lymph nodes was one (1.1 ± 2.5). The R0 resection rate was 89% with a tumor safety margin of 10.9 ± 14.3 mm (Appendix A).

Median disease-free survival was 18 months (95% CI (14.93–21.07)), and median overall disease-specific survival was 40 months (95% (33.54–46.46)) in patients with resected IHCC.

### 2.2. Distribution of Metastatic Lymph Nodes in Patients with Resected IHCC

Seven-hundred twenty-five patients with resected IHCC (63.7%) were noted to have retrieved lymph nodes. The number of retrieved lymph nodes was 5.9 ± 8.3. Among them, lymph node metastasis was observed in 293 patients (N1, 25.7%), and 432 patients (N0, 38.0%) were found to have no lymph node metastasis. Lymph node status could not be assessed in 413 patients (NX, 36.3%).

Analysis of topographical distribution of metastatic lymph nodes showed that the No. 12 lymph node (36%) was the most frequent metastatic node, and the No. 8 lymph node (21%) was the second most common metastatic node. (Appendix A).

In addition, left-sided IHCC was significantly associated with lymph node metastasis (right-sided, 112 of 338 patients (33.1%) vs. left-sided, 181 of 387 patients (46.8%), *p* < 0.001). Right-sided IHCC was related to the No. 12 lymph nodes (*p* = 0.065) and the No. 13 lymph nodes (*p* = 0.024, Table 2) (Appendix A).

### 2.3. Oncologic Impact of Metastatic Lymph Nodes in Patients with Resected IHCC

Metastatic lymph nodes showed adverse long-term oncologic impact in patients with resected IHCC. N1 patients (median 14 months, 95% CI (11.4–16.6)) showed worse disease-specific survival compared with NX patients (median 58 months, 95% CI (40.2–75.8), *p* < 0.001) and N0 patients (median 74 months, 95% CI (57.2–90.8), *p* < 0.001, Figure 1a).

A similar pattern of survival was also observed in the disease-free survival analysis. N1 patients (median disease-free survival 6 months, 95% CI (4.9–7.0)) showed inferior survival outcomes than the other two groups (Nx patients: median disease-free survival 25 months, 95% CI (15.6–34.4) and N0 patients: median disease-free survival 44 months, 95% CI (25.3–62.7), Figure 1b).

Interestingly, the number of metastatic lymph nodes was also significantly related to a negative oncologic impact in patients with resected IHCC. Patients with ≥4 metastatic lymph nodes (median 11 months, 95% CI (8.1–13.8)) showed the worst long-term oncologic outcome compared to those with one to three metastatic lymph nodes (median 19 months, 95% CI (14.4–23.6), *p* < 0.001) and zero metastatic lymph nodes (median 74 months, 95% CI (57.2–90.8), *p* < 0.001, Figure 1c).

### 2.4. Oncologic Impact of Number of Retrieved Lymph Nodes on N0, and N1 Patients with Resected IHCC

Among all patients with known lymph node status (*N* = 725 patients), the number of retrieved lymph nodes was not associated with significant survival differences in patients with resected IHCC (Appendix A).

However, the significant oncologic impact of the number of retrieved lymph nodes was observed according to lymph node metastasis status in patients with resected IHCC. In N0 patients, the number of retrieved lymph nodes was not associated with long-term survival differences (Figure 2a). However, in N1 patients, the number of retrieved lymph nodes was significantly associated with survival differences. Patients with one to three retrieved lymph nodes (median 13 months, 95% CI (10.4–15.6)) had poor survival outcomes compared to those with four to five retrieved lymph nodes (median 30 months, 95% CI (13.1–46.9), *p* = 0.045) and ≥six retrieved lymph nodes (median 14 months, 95% CI (10.9–17.1), *p* = 0.090). There were no statistically significant survival differences between patients with four to five retrieved lymph nodes and those with ≥six retrieved lymph nodes (*p* = 0.458, Figure 2b), suggesting that surgical effort to retrieve more than four lymph nodes will benefit patients with lymph node metastasis.

Analysis of the oncologic impact of the number of retrieved lymph nodes according to the number of metastatic lymph nodes clearly demonstrates that any effort to retrieve additional lymph nodes (four to five, vs. six<) does not provide a positive effect on the long-term survival of patients with more than four metastatic lymph nodes (median survival 9 months (95% CI: 0.0001–20.087) vs. median survival 11 months (95% CI: 8.283–13.717), *p* = 0.989), However, surgical effort to remove more than four (four to five, and six < vs. one to three) lymph nodes resulted in an improved long-term survival outcome in patients with less than four metastatic lymph nodes (median survival 30 months, (95% CI: 3.984–56.016), *p* = 0.016, and median survival 23 months, 95% (CI: 18.775–27.225), *p* = 0.001, vs. median survival 13 months (95% CI: 10.4–15.6), Figure 2c), suggesting that surgical retrieval of lymph nodes plays a significant oncologic role in some patients with limited lymph node metastasis.

### 2.5. Can Lymph Node Metastasis Be Preoperatively Predicted in Patients with Resected IHCC? Developing a Surgeon-Oriented Nomogram to Predict Lymph Node Metastasis

Among the detectable preoperative parameters that are thought be clinically available in usual clinical practice, age, symptoms, radiologic tumor size, planned operation, and preoperative tumor markers were considered to establish a statistical model to preoperatively predict lymph node metastasis in patients with IHCC (Table 3 and Figure 3). Logistic regression analysis identified symptoms at diagnosis (OR = 1.803 (95% CI: 1.245–2.612), *p* = 0.0018), operation type (for example, left extended hemihepatectomy, OR = 2.713 (95% CI 1.079–6.825), *p* = 0.0339), preoperative serum CEA level (OR = 1.966 (95% CI: 1.352–2.857), *p* = 0.0004), and preoperative serum CA 19-9 level (OR = 2.648 (95% CI: 1.837–3.819), *p* < 0.001) as independent predictors for lymph node metastasis.

The nomogram AUC was 0.737, and internal validation using 1000 bootstrap sampling from the primary cohort showed an AUC of 0.748 (95% CI: 0.706–0.788). These values suggest a stable accurate capacity for preoperatively predicting lymph node metastasis. In addition, the apparent performance of the nomogram to preoperatively predict lymph node status was tested in N0 and N1 patients. The calibration curve of the nomogram for the probability of LN metastasis demonstrated a good level of agreement between prediction and observation in the primary cohort (Hosmer and Lemeshow goodness of fit test, *p* = 0.4904, Appendix A).

### 2.6. Indirect External Validation of Nomogram in Nx Patients with Resected IHCC

The developed nomogram was applied in Nx patients with resected IHCC, whose lymph node status was not assessed, to estimate the discriminating capacity for predicting lymph node metastasis in patients with resected IHCC. Nx patients were categorized into two groups according to the calculated probability of lymph node metastasis. There were significant survival differences between Nx patients with low risk of lymph node metastasis (<50%) and those with high risk of lymph node metastasis (≥50%, *p* = 0.0002). When superimposing known survival outcomes of patients with resected IHCC according to lymph node metastasis, the survival outcomes of Nx patients with low risk of lymph node metastasis were similar to actual N0 patients (*p* = 0.0793, Bonferroni-corrected *p*-value = 0.5558). The survival outcomes of Nx patients with high risk of lymph node metastasis also showed no significant difference from that of known N1 patients (*p* = 0.5610, and Bonferroni-corrected [15] *p*-value > 0.99999, Figure 4). Therefore, in an effort to improve survival outcomes, Nx patients with high risk of lymph node metastasis should have undergone surgical management to retrieve at least four lymph nodes during hepatectomy.

### 2.7. Proposed Surgical Strategy in Lymph Node Management for IHCC

The following surgical strategy is proposed for managing lymph nodes during resection of IHCC. According to the calculated risk of lymph node metastasis based on the present surgeon-oriented nomogram, web-based and case-specific lymph node management can be suggested for surgical candidates with IHCC (http://40.121.207.11:8080/home3.jsp). This model calculates not only the risk of lymph node metastasis, but also provides a potential area of lymph node metastasis. In patients with a high calculated risk of lymph node metastasis (≥50%), surgical retrieval of at least four lymph nodes is recommended. Removal of different regional lymph nodes according to primary tumor location was considered, including the porta hepatis area (the most common location of metastatic lymph nodes), the No. 8 LNs (second most common location of metastatic lymph nodes), and the No. 7/perigastric LNs (common in left-sided IHCC). Conversely, patients with a low calculated risk of lymph node metastasis (<50%) only require lymph node sampling around the portahepatis for accurate tumor staging. In clinical practice, specimens can be sent for pathological evaluation as frozen-section biopsies for further decision making (Appendix A).

## 3. Discussion

The present Korean–Japan collaborative study to investigate the oncologic role of surgical effort to remove lymph nodes demonstrated that lymph node metastasis was strongly associated with poor long-term oncologic outcomes in resected IHCC. The present study emphasizes that not only lymph node metastasis status but also the number of metastatic lymph nodes should be considered when stratifying patients with resected IHCC and lymph node metastasis. Several validation studies are currently available that support the notion that the number of metastatic lymph nodes is more reliable and accurate for estimating long-term oncologic outcomes in resected pancreatic cancer [16,17,18,19] and gallbladder cancer [20,21].

Moreover, surgical retrieval of lymph nodes can have varying degrees of oncologic significance, according to the status of lymph node metastasis when managing IHCC. This observation is thought to be an important result of our study. There was no significant oncologic impact of the number of retrieved lymph nodes in resected IHCC *without* lymph node metastasis (N0). However, the number of retrieved lymph nodes provided a positive impact on the long-term survival in resected IHCC *with* lymph node metastasis (N1). The present study suggests that surgical effort to remove at least four lymph nodes is required in resection of IHCC with N1 to significantly impact long-term oncologic outcomes.

The long-term oncologic outcomes of patients with IHCC and lymph node metastasis are influenced not only by tumor biology, but also by surgical effort to remove regional lymph nodes. Any surgical effort to remove lymph nodes was not associated with improved survival outcome. Conversely, those with a limited number of metastatic lymph nodes (one to three) clearly demonstrated that surgical retrieval of at least four lymph nodes could provide a positive oncologic impact in resected IHCC. This observation may explain the controversial data on the issue of lymph node management in resected IHCC [22,23,24,25,26,27]. Studies enrolling many patients with ≥four metastatic lymph nodes may conclude that lymph node dissection does not play a significant role in managing IHCC.

In this study, a nomogram was developed to predict lymph node metastasis in resected IHCC based on preoperative detectable parameters because surgical management of lymph nodes can differ according to the presence of lymph node metastasis. Several studies have proposed nomograms for preoperatively predicting lymph node metastasis in resected IHCC. Meng et al. [28] proposed a nomogram based on preoperative serum CA 19-9, primary tumor site, measured lymph node size based on a CT scan, and tumor growth pattern. Huwan et al. [29] also proposed a radiomics nomogram for the preoperative prediction of lymph node metastasis.

It is more practical and applicable if a surgeon-oriented nomogram to preoperatively predict lymph node metastasis is proposed. The present nomogram is web-based operating system. Therefore, it can be used in every clinical office and even in operative theaters to provide additional information for appropriate decision-making regarding lymph node management for IHCC. Our proposed nomogram is simple and practical. Surgeons can apply the present nomogram to provide patients and their families with explanations on probable lymph node status even in the preoperative setting. To calculate the risk of lymph node metastasis, surgeons only need to know patient age, symptoms (incidental finding or symptomatic), radiologic tumor size (cm), the planned surgical extent of hepatectomy, preoperative serum CEA, and preoperative CA 19-9. This information can be obtained while interviewing patients in the preoperative setting and even during operation. Internal validation showed that this nomogram has an acceptable level of accuracy and predictive capacity to estimate lymph node metastasis status (AUC, 0.748, 95% CI: 0.704–0.788, and Hosmer and Lemeshow goodness of fit test, *p* = 0.4904), similar to two previous reports [28,29]. When the present nomogram was applied in 413 patients whose lymph node status was not assessed (Nx patients), patients were categorized into two groups with discrete long-term survival outcome according to the calculated risk of lymph node metastasis. In turn, the long-term oncologic outcomes of Nx patients were exactly superimposed on those of patients with known lymph node metastasis. Our study emphasizes that Nx patients with a high calculated risk of lymph node metastasis (≥50%) could obtain oncologic benefit from surgical retrieval of at least four lymph nodes during curative resection.

This study is based on the largest study populations of resected IHCC. However, there are several limitations. First, the study design is retrospective, so it is difficult to identify the exact location of retrieved LNs, the exact extent of LN dissection, and the cN-stage to improve the accuracy for predicting metastatic LNs. According to the present review of topographic location of metastatic lymph nodes, the No. 12 and No. 8 lymph nodes were the most frequent occurring nodes. It is well known that regional lymph nodes differ according to the location of the primary IHCC [30]. Systemic lymph node dissection according to tumor location, including the No. 12 and No. 8 areas, is suggested to obtain more than four lymph nodes when lymph node metastasis is highly suspected by nomogram or by intraoperative frozen section biopsy of suspicious enlarged lymph nodes. Secondly, a fairly large number of patients (*N* = 413, 36.3%) were noted to have Nx (the amount of harvested lymph nodes and amount of positive lymph nodes could not be assessed), which could potentially introduce bias in the study. This may be a result of different surgical interventions towards LNs (surgeon bias), and non-standardized pathologic evaluation methods (pathologist bias). In this study, we did not exclude Nx patients because this group was used for only indirect validation. In addition, Nx is categorized as one of the N staging systems (Nx, N0, and N1), and Nx can definitely occur in our clinical practice. However, for further in depth analysis of the oncologic role of LNM in resected IHCC, a well designed prospective study will be necessary. Lastly, the accuracy of the present nomogram was tested by only internal validation. Instead, the potential capacity of the proposed nomogram to predict lymph node metastasis was indirectly shown in Nx patients with resected IHCC, not by external validation using a different cohort of patients. Therefore, a well-designed external validation study based on prospective study protocol should be performed in the near future to strengthen the value of the proposed nomogram-based case-specific LN management strategy in resected IHCC.

## 4. Materials and Methods

Study design and data collection: This study was approved by the Korea Association of Hepato-Biliary-Pancreatic Surgery and the Japanese Society of Hepato-Biliary-Pancreatic Surgery as an international multicenter collaborative study. Patients with resected IHCC were selected from digitalized patient records of individual institutions who agreed with the present international collaborative study protocol. From January 2000 to December 2011, the medical records of patients who underwent potential surgical resection for IHCC were retrospectively reviewed. The clinical and pathological parameters, including the number of retrieved lymph nodes, metastatic lymph nodes, and long-term survival data, were collected from medical records of individual patients to establish the present data-base for the Korea–Japan collaborative study. This study was approved by the Institutional Review Board of Yonsei University College of Medicine (IRB# 4-2014-0850).

Statistics: All data were collected and analyzed at the Department of Surgery and Department of Biostatistics, Yonsei University College of Medicine, Seoul, Korea. Continuous variables were described as mean ± standard deviation, and categorical variables were described as frequency (%). Student’s *t*-test, chi-square test, or Fisher’s exact test were performed. Survival was assessed with Kaplan–Meier analyses. Survival outcomes were compared using log–rank tests with 95% confidence interval (CIs) to identify associations between clinical factors and survival outcomes.

When developing the prediction model, the Youden Index [31] was used to determine the optimal cutoff points for preoperative serum CA19-9 and CEA. Preoperatively available clinical parameters that achieved statistical significance in the univariable logistic regression model and were considered of clinical importance were included in the final model. A nomogram was developed by using the package *rms* in R version 3.1.3 (http://www.r-project.org/) on the basis of the results of the multivariable logistic regression model. Internal validation was performed using 1000 bootstrapped resamples, which were also used to generate the calibration plot. To quantify the discriminative ability of the final model, AUC was measured. Calibration was evaluated by calibration plot, a graphic representation of the relationship between observed and predicted probability. The Hosmer–Lemeshow goodness-of-fit test was performed [32]. The agreement between predicted and observed probability was assessed, and *p*-value > 0.05 indicates good calibration. To further assess the validation, Kaplan–Meier curves of Nx patients were plotted according to calculated risk of lymph node metastasis by nomogram over risk stratified groups by actual lymph node metastasis. *p*-values < 0.05 were considered statistically significant. To strictly validate the oncologic significance of the calculated risk of lymph node metastasis, Bonferroni-corrected *p*-values were applied when comparing actual survival.

## 5. Conclusions

In summary, the present Korea–Japan collaborative study suggests a potential strategy to tailor surgical approaches based on the calculated risk of lymph node metastasis when treating IHCC. Patients with high risk of lymph node metastasis should be treated by lymph node dissection to obtain more than four lymph nodes during IHCC resection. Patients with a low risk of lymph node metastasis should simply be managed by margin-negative hepatectomy with lymph node sampling to avoid unnecessary lymph node management and to ensure accurate staging. This surgical concept should be approved by a multi-national and multi-center randomized control study with reasonable consensus-based study protocol. Future research on this topic is needed.

## Figures and Tables

**Figure 1 cancers-13-00445-f001:**
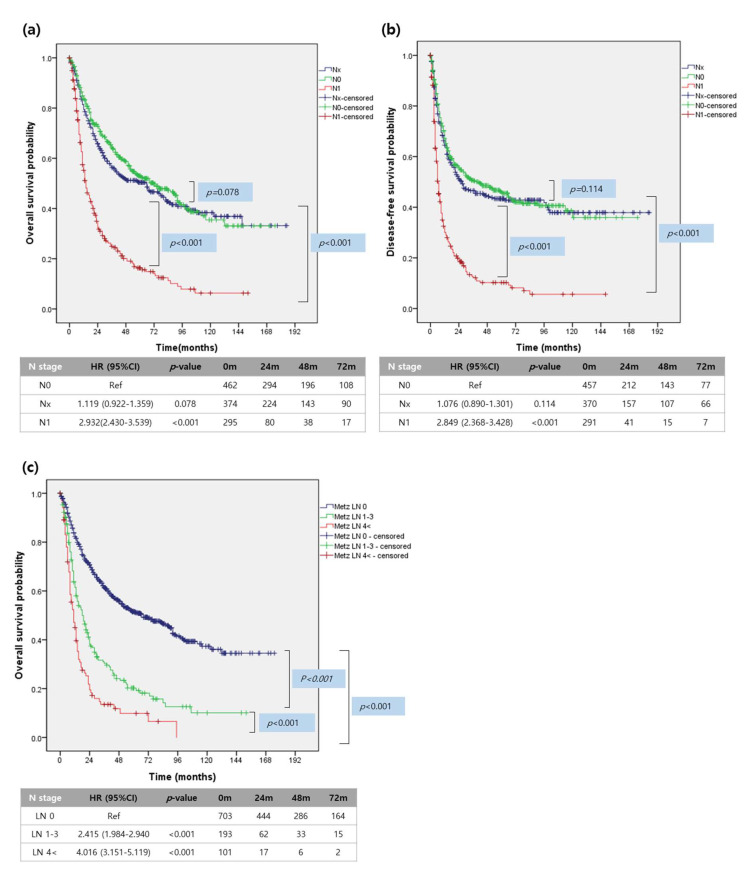
Survival analysis of resected IHCC according to N-stage. (**a**) Overall survival survival of resected IHCC (**b**) Disease-free survival analysis. (**c**) Overall survival according to the number of metastatic lymph nodes.

**Figure 2 cancers-13-00445-f002:**
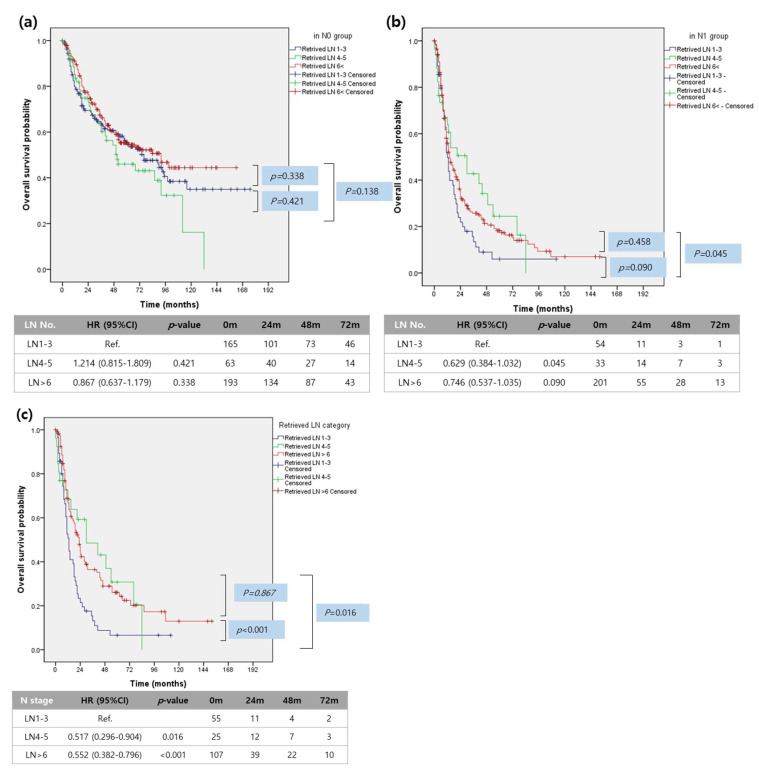
Survival analysis according to number of retrieved lymph nodes. (**a**) Overall survival analysis according to the number of retrieved lymph nodes in the N0 group. (**b**) Overall survival analysis according to the number of retrieved lymph nodes in the N1 group. (**c**) Overall survival analysis according to the number of retrieved lymph nodes in patients with less than 4 (1–3) metastatic lymph nodes.

**Figure 3 cancers-13-00445-f003:**
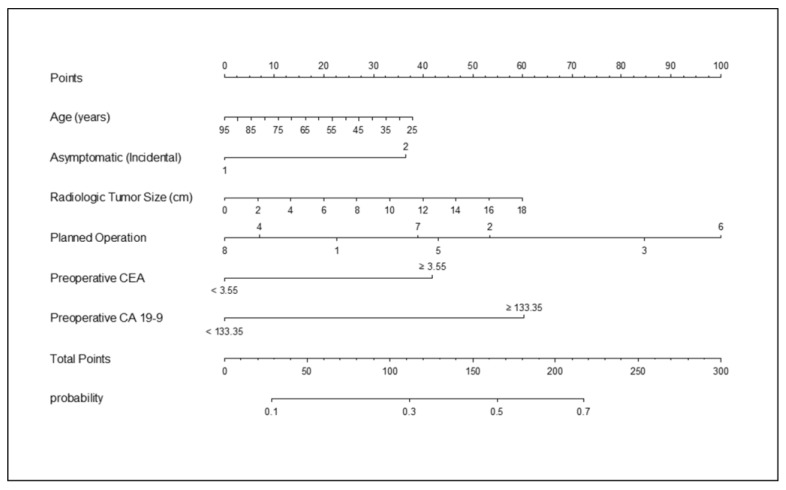
Nomogram developed to preoperatively predict lymph node metastasis in patients with IHCC. Presence/absence of symptoms, one = no symptoms, two = symptomatic; planned operation, one = left lateral segmentectomy, two = left hemihepatectomy, three = left extended hemihepatectomy, four = right hemihepatectomy, five = right extended hemihepatectomy, six = trisectionectomy, seven = bisegmentectomy, eight = segmentectomy.

**Figure 4 cancers-13-00445-f004:**
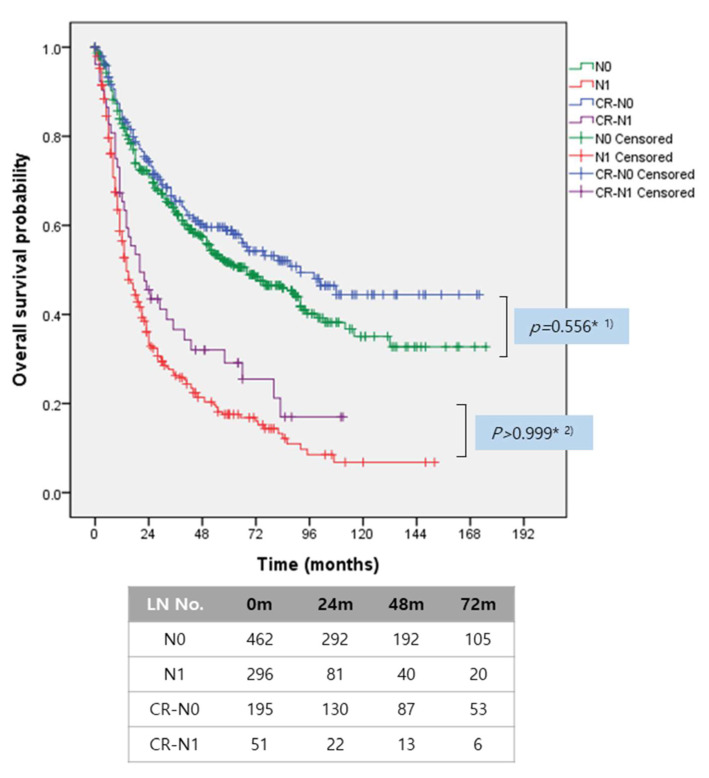
Correlation of survival outcomes between actual lymph node status and calculated risk of lymph node metastasis in resected IHCC. Survival outcomes according to calculated risk of lymph node metastasis were similar to those according to actual lymph node metastasis, (1) Bonferroni corrected *p*-value = 0.5558, (2) Bonferroni-corrected *p*-value > 0.9999, CR, calculated risk.

**Table 1 cancers-13-00445-t001:** Demographics of patients with resected intrahepatic cholangiocarcinoma (IHCC).

	Patients (*N* = 1138)
Sex	
Male	758 (66.6%)
Female	380 (33.4%)
Age (years)	63.4 ± 9.8
Previous symptoms	
Yes	535 (47.0%)
No	603 (53.0%)
American Sosciety of Anesthesiologists (ASA) score	
1	250 (24.6%)
2	662 (65.2%)
3	98 (9.6%)
4	1 (0.1%)
5	1 (0.1%)
Karnofsky scale	
50	1 (0.1%)
60	5 (0.5%)
70	25 (2.6%)
80	142 (14.8%)
90	518 (54.1%)
100	267 (27.9%)
Tumor location side	
Right liver	558 (50.4%)
Left liver	549 (49.6%)
Operation name	
Lt. lateral sectionectomy	75 (6.6%)
Lt hemihepatectomy	325 (28.6%)
Lt extended hepatectomy	126 (11.1%)
Rt hemihepatectomy	295 (25.9%)
Rt extended hepatectomy	79 (6.9%)
Trisectionectomy	31 (2.7%)
Bisegmentectomy	65 (5.7%)
Segmentectomy	142 (12.5%)
Lymph node retrieval	
No	413 (36.3%)
Yes	725 (63.7%)
Serum Carcinoembryonic antigen (CEA) (ng/mL)	21.5 ± 111.3
Serum 19-9 (U/mL)	2273.9 ± 10,816.4

**Table 2 cancers-13-00445-t002:** Topographical relationship between metastatic lymph nodes and primary tumor location of resected IHCC.

Tumor Side	Right-Sided (*N* = 112)	Left-Sided (*N* = 181)	*p*
No. 7 Lymph node (LN)			0.013
No	109 (97.3%)	162 (89.5%)	
Metastasis	3 (2.7%)	19 (10.5%)	
No. 12 LN			0.065
No	47 (42.0%)	96 (53.0%)	
Metastasis	65 (58.0%)	85 (47.0%)	
No. 13 LN			0.024
No	93 (83.0%)	166 (91.7%)	
Metastasis	19 (17.0%)	15 (8.3%)	
Perigastric LN			0.052
No	109 (97.3%)	166 (91.7%)	
Metastasis	3 (2.7%)	15 (8.3%)	

**Table 3 cancers-13-00445-t003:** Univariate and multivariate analysis for preoperatively predicting lymph node metastasis in resected IHCC.

Variable	Univariate		Multivariate	
	OR (95% CI)	*p*-Value	OR (95% CI)	*p*-Value
Sex (Male/Female)	1.08 (0.798–1.463)	0.6164		
Age, years	0.989 (0.974–1.004)	0.1401	0.991 (0.974–1.009)	0.3427
Chief complaint (no/yes)	1.81 (1.344–2.438)	0.0001	1.803 (1.245–2.612)	0.0018
ASA	0.727 (0.505–1.046)	0.0854		
	0.873 (0.479–1.591)	0.6575		
Karnofsky score	0.995 (0.975–1.016)	0.657		
Radiologic tumor size, cm	1.092 (1.033–1.156)	0.0021	1.055 (0.982–1.134)	0.1415
Gross type	0.917 (0.748–1.124)	0.4023		
Tumor location (Right/Left)	1.546 (1.148–2.082)	0.0042		
Number of the tumor	1.303 (0.985–1.723)	0.064		
Left hemihepatectomy	1.122 (0.555–2.268)	0.7495	1.646 (0.685–3.952)	0.2649
Left extended hemihepatectomy	1.608 (0.755–3.425)	0.2185	2.713 (1.079–6.825)	0.0339
Right hemihepatectomy	0.651 (0.315–1.343)	0.2448	0.777 (0.319–1.896)	0.5799
Right extended hemihepatectomy	1.004 (0.44–2.288)	0.9934	1.39 (0.515–3.752)	0.5156
Trisectionentectomy	2.514 (0.804–7.862)	0.1129	3.488 (0.925–13.15)	0.0651
Bisegmentectomy	0.933 (0.365–2.384)	0.8854	1.301 (0.411–4.117)	0.6539
Segmentectomy	0.304 (0.121–0.766)	0.0116	0.694 (0.233–2.066)	0.5119
Preoperative CEA	2.113 (1.537–2.903)	<0.0001	1.966 (1.352–2.857)	0.0004
Preoperative CA 19-9	3.389 (2.475–4.643)	<0.0001	2.648 (1.837–3.819)	<0.0001

## Data Availability

The data presented in this study are available on request from the corresponding author. The data are not publicly available due to IRB restriction regarding personal information.

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
