# Peer review of "Should Lymph Nodes Be Retrieved in Patients with Intrahepatic Cholangiocarcinoma? A Collaborative Korea–Japan Study"

_cancers, 2021, doi:10.3390/cancers13030445_

Round 1

Reviewer 1 Report

Great Job and very Interesting. 

I think it is better to write the meaning of the acronyms also in the abstract text (for example LN: lymphnodes (LN) ... etc). For the rest: the contents, the exhibition, in short, all the work appears commendable

Author Response

Thank you very much for reviewer's comments and suggestions.

We attached file for authors' responsed to reviewer's comments (R1) to this system.

Reviewer 2 Report

The authors present a retrospective series which aimed to identify predictors of lymph node metastasis in patients with intrahepatic cholangiocarcinoma (IHCC). Based on the nomogram, the authors propose a surgical strategy including lymphadenectomy of >4 lymph nodes in patients with a high risk of lymph node metastasis, and a more conservative approach in low risk patients. In addition, the authors suggest that future validation is required in prospective series.

I believe that the study is interesting and methodologically sound. In general, the manuscript requires only minor revision of spelling and grammar. The model performs well, with an AUC of .748. I have some further suggestions to improve the manuscript, which can be found below:

Major comments:

  • In a fairly large amount of patients (n=413, 36.3%), the amount of harvested lymph nodes and amount of positive lymph nodes could not be established. What was the reason for this amount of missing data? Were this data deemed missing at random? This could potentially introduce bias in the study, especially since this population was used as an internal validation cohort. This should be added to the limitations section.
  • How were patients selected for inclusion? Did the authors perform a query on digital patient records to identify patients with IHCC? Was it part of national registries, or were previous retrospective databases used? Please include in the Methods section.
  • It would greatly strengthen the paper if external validation could be performed, but I assume it can be challenging to acquire an international validation cohort.
  • Were differences observed in outcome between centers/countries? It could be interesting to perform a sensitivity-analysis, to observe the impact of country of origin. Also, the baseline characteristics of the derived cohorts per country could be compared. Please also add number of centers, and whether these centers were considered high, medium or low surgical volume centers.
  • Was radiologic N-stage (suspected lymph node metastasis on pre-operative imaging) included in the univariate screen? It can be assumed that patients with a radiologic suspicion of N1-disease have a higher likelihood of indeed having lymph node metastasis.

Minor comments:

  • Use of abbreviations i.e. ICC (line 68)/IHCC should be consistent throughout the manuscript. Please review the paper and adjust if needed.
  • It would make sense to remove decimals from number of resected lymph nodes (line 104), since in general lymph nodes are not partially removed.
  • Odd’s ratios and p-values could be added to the abstract, instead of just mentioning a significant correlation between factors and oncological outcome.
  • Was pathological examination of the resection specimen standardized between centers, i.e. could there be a bias between centers based on the technique to assess the pathological specimen?

Author Response

Thanks you very much.

We attached file for authors' response to reviewer comments (R2) to this system.

Reviewer 3 Report

The work by Kang et al. was focused on the study of the contribution of lymph node metastasis to the prognosis of intrahepatic cholangiocarcinoma (iCCA). The authors conducted a collaborative Korea-Japan study in which they analyzed the medical records of patients with iCCA, measuring the short- and long-term oncological outcomes according to the presence of lymph node metastasis (LNM). In this study, a total of 1138 patients with iCCA were enrolled and 413 of these patients underwent lymph node management. Importantly, 293 patients had LNM.  Importantly, the No.12 LN was the most frequent metastatic node followed by No.8 LN. LNM resulted in adverse long-term oncologic impact in patients with resected iCCA and the increased number of LNM was related with worse overall and disease-free survival. Additionally, the authors also found that surgical retrieval of more than 4 LNs could improve the survival outcome in these patients. Finally, the authors constructed a nomogram based on several simple clinical findings that allowed to predict the presence of LNM with a good AUC value (0.748). The manuscript is very well-written and the findings herein presented are extremely sound and relevant. The analysis were carefully conducted and I only have small comments to this excellent work.

  1. In introduction, I missed a sentence clearly explaining the aim of this study. Please consider to add one.
  2. In page 2, in line 68, please correct ICC by IHCC.
  3. In the Kaplan-meyer curves, it would be pivotal to add the number of patients at risk in each of the time points evaluated. Also, please calculate de hazard ratios (HRs) for each one of the conditions.
  4. Regarding the prediction model, does it also help in the prediction of the number of metastatic sites?   

Author Response

Thank you very much.

We attached file of authors responses to reviewer' comments (R3) to this system.

Round 2

Reviewer 2 Report

The authors have improved their manuscript according to the aforementioned suggestions. I have no further comments and recommend publication of the paper.